# Determination of Kinetic Parameters and Identification of the Rate-Determining Steps in the Oxygen Exchange Process for LaNi_0.6_Fe_0.4_O_3−*δ*_

**DOI:** 10.3390/ijms241613013

**Published:** 2023-08-21

**Authors:** Natalia Porotnikova, Dmitriy Zakharov, Anna Khodimchuk, Edhem Kurumchin, Denis Osinkin

**Affiliations:** 1Laboratory of Kinetics, The Institute of High-Temperature Electrochemistry of the Ural Branch of the Russian Academy of Sciences, 20 Akademicheskaya Street, Yekaterinburg 620137, Russia; stall92@mail.ru (D.Z.); annlocked@gmail.com (A.K.); e.kurumchin@ihte.uran.ru (E.K.); osinkinda@mail.ru (D.O.); 2Laboratory of Electrochemical Devices and Fuel Cells, Ural Federal, University Named after the First President of Russia B. N. Yeltsin, 19 Mira Street, Yekaterinburg 620002, Russia; 3Department of Environmental Economics, Graduate School of Economics and Management, Ural Federal University Named after the First President of Russia B. N. Yeltsin, 19 Mira Street, Yekaterinburg 620002, Russia

**Keywords:** LaNi_0.6_Fe_0.4_O_3−*δ*_, rate-determining step, oxygen diffusion, surface, electrode material

## Abstract

The mixed ionic and electronic oxide LaNi_0.6_Fe_0.4_O_3−*δ*_ (LNF) is a promising ceramic cathode material for solid oxide fuel cells. Since the reaction rate of oxygen interaction with the cathode material is extremely important, the present work considers the oxygen exchange mechanism between O_2_ and LNF oxide. The kinetic dependence of the oxygen/oxide interaction has been determined by two isotopic methods using ^18^O-labelled oxygen. The application of the isotope exchange with the gas phase equilibrium (IE-GPE) and the pulsed isotope exchange (PIE) has provided information over a wide range of temperatures (350–800 °C) and oxygen pressures (10–200 mbar), as each method has different applicability limits. Applying mathematical models to treat the kinetic relationships, the oxygen exchange rate (*r_H_*, atom × cm^−2^ × s^−1^) and the diffusion coefficient (*D*, cm^2^/s) were calculated. The values of *r_H_* and *D* depend on both temperature and oxygen pressure. The activation energy of the surface exchange rate is 0.73 ± 0.05 eV for the PIE method at 200 mbar, and 0.48 ± 0.02 eV for the IE-GPE method at 10–20 mbar; for the diffusion coefficient, the activation energy equals 0.62 ± 0.01 eV at 10–20 mbar for the IE-GPE method. Differences in the mechanism of oxygen exchange and diffusion on dense and powder samples are observed due to the different microstructure and surface morphology of the samples. The influence of oxygen pressure on the ratio of contributions of different exchange types to the total oxygen exchange rate is demonstrated. For the first time, the rate-determining step in the oxygen exchange process for LNF material has been identified. This paper discusses the reasons for the difference in the mechanisms of oxygen exchange and diffusion.

## 1. Introduction

Binary and complex oxide materials have attracted much attention in the last decade due to their widespread use in photosensors, light-emitting diodes, electronic goods, glass, and other optoelectronic devices in modern industry, medicine, and agriculture [1,2,3,4,5,6,7]. Complex oxide materials with mixed oxygen-ion and electron conductivity have attracted considerable research attention due to their practical applications such as steam reforming, partial oxidation for the conversion of natural gas to synthesis gas, and materials for electrochemical applications [8,9,10].

Fuel cell development is an exciting field both environmentally and technologically due to the possibility of direct and efficient conversion of chemical energy into electrical energy. To lower production costs, reducing operating temperatures is an important task in scaling up technology developments. One of the most important issues in solid oxide fuel cell (SOFC) development [11] is improving electrode performance, particularly on the cathode side. Recently, researchers have focused considerably on exploring the use of doped perovskites as materials for solid oxide fuel cells (SOFCs) [12,13,14]. LaNi_1−*x*_Fe*_x_*O_3−*δ*_ has attracted interest due to its resistance to certain degradation effects, so these materials may find application as oxygen electrodes for future generations of SOFCs with lower degradation rates and longer lifetimes [15].

It is known that in the LaNi_1−*x*_Fe*_x_*O_3−*δ*_ series, the composition with an iron content *x* = 0.4 has the highest electronic conductivity [16]. The oxygen non-stoichiometry (*δ*) and the thermal and chemical expansion (Δ*L*/*L*_0_) are dependent on the temperature and oxygen pressure. The values of *δ* and Δ*L*/*L*_0_ for LaNi_0.6_Fe_0.4_O_3−*δ*_ were determined in [17]. Positive features of the LaNi_0.6_Fe_0.4_O_3−*δ*_ material are the absence of cobalt and strontium in the composition, which inhibit degradation processes [18,19]; a satisfactory coefficient of thermal expansion ((11.4–13) × 10^−6^ K^−1^ [15,20,21]) close to electrolytes Ce_0.9_Gd_0.1_O_2–*δ*_ (12.5 × 10^−6^ K^−1^ [22]), BaCe_0.8_Y_0.2_O_3−*δ*_ (11.6 × 10^−6^ K^−1^ [23]), and La_0.9_Sr_0.1_Ga_0.8_Mg_0.2_O_3−*δ*_ (11.4 × 10^−6^ K^−1^ [24]); and the absence of chemical interaction with chromium-containing steels [25]. The electron conductor LaNi_0.6_Fe_0.4_O_3−*δ*_ can effectively function as a current collector for bilayer electrodes containing functional layers composed of layered nickelate [26,27,28,29] or composite electrodes [30,31,32]. The method of synthesis greatly affects the dispersibility, morphology, particle size, and sinterability of the material. The nitrate combustion method and the modified Pechini method demonstrate superior performance in forming ceramic layers compared to the solid phase preparation method [33]. The collector layers that are based on LaNi_0.6_Fe_0.4_O_3−*δ*_ demonstrate the most isotropic microstructure among all the layers, with small particles and a well-developed porous structure. Such a microstructure favors the current distribution throughout the layer and consequently results in an enhancement of the electrode’s electrochemical activity.

Investigating the kinetics of oxygen exchange and diffusion in mixed conductivity solid oxides, like the LaNi_1−*x*_Fe*_x_*O_3−*δ*_ complex oxides, is critical since the rates of these processes affect fundamental parameters required for synthesizing these materials and operating high-temperature electrochemical devices, which employ them as electrode base materials. Several papers [34,35,36] have determined oxygen exchange rates and diffusion coefficients. Nonetheless, a comprehensive investigation of the mechanisms and rate-determining steps for these processes still requires clarification. This study aims to investigate the interaction mechanism of gaseous oxygen with LaNi_0.6_Fe_0.4_O_3−*δ*_ (LNF) using two independent oxygen isotope exchange methods: the pulse isotope exchange (PIE) and the isotope exchange with the gas phase equilibration (IE-GPE). These techniques allow the determination and comparison of kinetic parameters in equilibrium conditions at high and low oxygen partial pressures. The findings will determine the nature of the rate-determining step of oxygen exchange.

## 2. Results and Discussion

LNF powder has a perovskite structure with rhombohedral distortions, as evidenced by the X-ray diffraction data in Figure 1a. The space group is defined as *R*-3*c* (#167), and the structure belongs to the rhombohedral symmetry class. The lattice parameters are as follows: *a* = *b* = 5.5064(3) Å, *c* = 13.2586(8) Å, *V* = 348.15(4) Å^3^, *α* = *β* = 90°, and *γ* = 120°. The X-ray reflections recorded can be fully explained by LNF structure. The powder’s specific surface area was measured to be 5.6 ± 0.1 m^2^/g. Figure 1b shows the size distribution function of the powder particles. The particles have an average size of about 3 μm, with the majority of particles being this size. There is some agglomeration, resulting in a few larger particles that are up to 40 μm in size.

The microstructure, elemental composition, and quantitative content of the ceramics were assessed using scanning electron microscopy with energy-dispersive X-ray microanalysis. The results are shown in Figure 2. The secondary electron (SE) image provides detailed information about the plate’s polished surface. The ceramics have a polished surface with depressions caused by the roughness of coarse agglomerated particles. The chemical and elemental composition is homogeneous, which is validated by the La, Ni, and Fe distribution maps. The surface topography is visible in the backscattered electron (BSE) image. The size of closed near-surface defects and pores ranges from 1.8 to 9 μm. The ceramic grains are agglomerate and have individual sizes of 2–5 μm.

Elemental concentrations were analyzed in three different segments using energy-dispersive X-ray microanalysis. Details of the elemental content are provided in Table 1. Elemental concentrations show slight differences in the three comparisons and have been used to enhance the chemical composition, which can be represented as LaFe_0.38_Ni_0.62_O_2.75_.

Certified materials were used to measure the kinetics of oxygen isotope exchange between the gas phase and the oxide. Figure 3a presents the temperature dependencies of the mole fractions of oxygen isotopologues and the mean ^18^O fraction in the gas phase (*α* = 0.5 × *x*_34_ + *x*_36_) at a gas carrier flow rate of 3.6 L/h.

Figure 3a displays sigma-shaped dependencies for *x*_32_, *x*_34_, *x*_36_, and *α*. In a paper [37], Bouwmeester H.J.M. found that the temperature dependencies of *x*_34_ for YSZ and La_2_NiO_4±*δ*_ materials have a Gaussian-like shape. Our temperature dependence analysis of *x*_34_ shows no extremum within the studied temperature range. The absence of an extremum indicates a high activity of LNF in oxygen surface exchange during PIE experiments. The catalytic activity is significant, resulting in the rapid consumption of both ^16^O^18^O and ^18^O_2_ from the pulse by LNF bulk. The result is only the formation of ^16^O_2_, and consequently, there is a full consumption of the pulses at temperatures above 600 °C, for both 3.6 and 4 L/h gas carrier flows. Due to the high activity of LNF powder at temperatures above 600 °C, it is impossible to analyze the oxygen surface exchange mechanism at these temperatures using the PIE method. The study of the mechanism of oxygen surface exchange was carried out at temperatures above 600 °C using dense ceramic material with the IE-GPE method.

To describe the mechanism of oxygen surface exchange from PIE results, the three types of exchange models were applied [38]. The model suggests that the kinetics of oxygen isotope exchange between the gas phase and a solid can be described by three independent kinetic parameter types of exchange. The types of exchange correspond to the number of oxygen atoms that are exchanged between oxygen isotopologues in the gas phase, namely ^16^O_2_, ^16^O^18^O, ^18^O_2_, and a solid, through one elementary act of exchange. These acts can involve zero (*r*_0_-type), one (*r*_1_-type), or two (*r*_2_-type) atoms [39], according to Equations (1)–(3), where O*_s_* refers to the oxygen in the crystal lattice of the oxide.
(1)O162+O182↔r02O16O18,
(2)O16O18+O16S↔r1O162+O18S,
(3)O182+2O16S↔r2O162+2O18S.

Ezin A.N. et al. [40] describe a mathematical model that allows for a quantitative analysis of three types of oxygen transport: *r*_0_, *r*_1_, and *r*_2_.

Oxygen dissociative adsorption (*r*_a_) and incorporation (*r*_i_) rates were calculated by fitting the results of fitting with three types of exchange models using follow constraint equations:(4)$$r_{0}=r_{a}{\left(1-\frac{r_{i}}{r_{a}+r_{i}}\right)}^{2},$$
(5)$$r_{1}=\frac{2r_{a}r_{i}}{r_{a}+r_{i}}\left(1-\frac{r_{i}}{r_{a}+r_{i}}\right),$$
(6)$$r_{2}=r_{a}{\left(\frac{r_{i}}{r_{a}+r_{i}}\right)}^{2}.$$

The system of isotope–kinetic differential equations for the three types of exchange model can be written as follows:(7)$$\{\begin{array}{c} \frac{d\alpha }{d\tau }=r_{H}\left(\alpha_{S}-\alpha \right)\\ \frac{dY}{d\tau }=-rY+r_{2}{\left(\alpha_{S}-\alpha \right)}^{2} \end{array},$$
where *α_s_* means ^18^O fraction in a solid; *Y* is a variable represented by a residual between *x*_34_, expressed with the binomial distribution of ^18^O in O_2_ and *x*_34_ from the experiment (*Y* = 2*α*(1−*α*) − *x*_34_); *r_H_* is the oxygen heterogeneous exchange rate (*r_H_* = 0.5*r*_1_ + *r*_2_); and *r* is the total rate of oxygen isotope exchange between the gas phase and the solid (*r* = *r*_0_ + *r*_1_ + *r*_2_) [39].

The ^18^O fraction was calculated using the relationship:(8)$$\alpha =0.5x_{34}+x_{36},$$
where α is the ^18^O fraction during the different interaction time and *x*_34_ and *x*_36_ are the concentrations of ^16^O^18^O and ^18^O_2_ in the gas phase, respectively. The interaction time dependencies of *α* were determined by the equation, where τ is the time of interaction, *α*^0^ is the ^18^O fraction at τ = 0, γ is the ^18^O fraction at τ = ∞, and *λ*λ is the ratio between the number of oxygen atoms in the gas phase and a solid.
(9)$$\alpha =\gamma +\left(\alpha^{0}-\gamma \right)e^{-r_{H}\left(1+\lambda \right)\tau },$$

The analytical solution of system (9) is written as:(10)$$Y=\left[\frac{2r_{2}{\left(1+\lambda \right)}^{2}{\left(\alpha^{0}-\gamma \right)}^{2}}{r-2r_{H}\left(1+\lambda \right)}\left(e^{\left[r-2r_{H}\left(1+\lambda \right)\right]\tau }-1\right)+Y^{0}\right]e^{-r\tau }.$$

The rates of the three types of exchange, the heterogeneous exchange rate, and the total oxygen isotope exchange rate were determined by fitting the time dependencies of *α* and *Y* interaction with the analytical solution of the system (7). The same rate values were used for both *α* and *Y* dependencies. The experimental data are presented by six points: three points for α and three points for *Y*. These points correspond to three different interaction times: 0 s (for the isotopic composition of O_2_ before the pulse), 0.9 s (for the isotopic composition of O_2_ after the pulse at a flow rate of 4 L/h), and 1 s (for the isotopic composition of O_2_ after the pulse at a flow rate of 3.6 L/h). The as-calculated values of *α* and *Y* at three different values of interaction times fitted with the three types of exchange model are presented in Figure 3b as points and lines, respectively.

To determine the rates of the three types of exchange (*r*_0_, *r*_1_, and *r*_2_), the oxygen exchange rate (*r_H_*), and the total oxygen isotope exchange rate (*r*) using the IE-GPE method, the same approach as used in the PIE method was followed. The difference between the PIE and IE-GPE methods lies in the number of experimental points collected. In contrast to the PIE experiments, the IE-GPE experiments were carried out with continuous sampling of small gas portions from a closed reaction chamber with the sample to the mass spectrometer. Figure 4 demonstrates the time dependencies of oxygen isotopologues and ^18^O fractions that were measured using the IE-GPE method.

The application of IE-GPE allows us to collect comprehensive profile data on isotope exchange kinetics and calculate both the oxygen exchange rate (*r_H_*, atom × cm^−2^ × s^−1^) and the oxygen diffusion coefficient (*D*, cm^2^ × s^−1^). However, IE-GPE can only be applied at relatively low partial pressures of about 10–20 mbar, whereas the PIE data can give us information about oxygen surface exchange kinetics at higher *P*_O_2__, 200 mbar. To compare the data obtained by two different methods, the values of the isotope exchange rates and the rates of the individual steps of dissociative adsorption and oxygen incorporation were calculated as a function of the sample area and the numbers of O atoms in the gas phase [*r_H_*, *r_a_* or *r_i_*, atom×cm^−2^ × s^−1^]. And, to compare the isotope exchange rates of our results with data reported in the literature, the surface exchange coefficient [*k*, cm × s^−1^] was used, which was calculated as follows:(11)$$k=\frac{r_{H}M}{\left(3-\delta \right)N_{A}\rho },$$
where *ρ* is the crystallographic density of LNF; *M* is the molar weight of LNF; *δ* is an oxygen non-stoichiometry in LNF, and *N_A_* is Avogadro’s constant. The following section will discuss the results of the diffusion coefficient and oxygen exchange coefficient obtained by processing the data from the IE-GPE method.

Figure 5 shows the temperature dependence of the oxygen diffusion coefficients measured by the IE-GPE method, which is compared with the oxygen diffusion coefficients measured by Nishi M. et al. [34] and Budiman R.A. et al. [35] using secondary ion mass spectrometry (SIMS). One can note that the oxygen pressure in our experiments and literature is not identical; these differences vary by order of magnitude. Moreover, the IE-GPE and SIMS methods have different approaches to data acquisition (in situ in the case of IE-GPE vs. ex situ in the case of SIMS). Details of the experimental conditions and methods are given in Table 2.

Our data for the oxygen diffusion coefficient at 800 °C have the smallest deviation; the values differ by a factor of two. With the decrease in temperature, the discrepancy between our data and the literature data increases, so that at 600 °C our results exceed the literature values by two orders of magnitude. The activation energy values, which can be found in Table 2, are also different. According to our calculations, the effective activation energy is significantly lower. It can be assumed that the discrepancies are likely caused by the effect of grain boundary diffusion. The IE-GPE method can only provide an estimation for the average oxygen diffusion coefficient in both bulk and grain boundaries based on Klier’s model. Isotopic profiling methods provide diffusion coefficients for oxygen in the bulk grains. In contrast to the IE-GPE method, the SIMS method is a local method that studies oxygen diffusion in separate grains of a specimen through layer-by-layer analysis using a beam diameter of a few microns. Thus, the difference between the diffusion coefficients could be related to different types of diffusion averaging and single grain measurements using these methods.

The diffusion transport slows down when there is a change of between 10 and 20 mbar (Figure 5). Similar behavior can be observed in perovskite oxide materials that follow the *D* = *f*(*P*_O_2__) relationship [41,42,43]. As LaNi_0.6_Fe_0.4_O_3−*δ*_ oxide has an oxygen transfer vacancy mechanism, decreasing oxygen pressure results in a decrease in oxygen non-stoichiometry, which subsequently slows down diffusive oxygen transfer.

LNF oxide exhibits mixed oxygen-ionic and electronic conductivity, with electron transport prevailing. There are no direct measurements of the ionic conductivity of this oxide in the literature due to electron transport interfering with the separation of ionic transport. The diffusion coefficient data, obtained using the in situ IE-GPE method, can be used to estimate the oxygen-ionic conductivity directly by applying the Nernst–Einstein equation:(12)$$\sigma =D\frac{nz^{2}e^{2}}{k_{B}T},$$
where *D* is the oxygen diffusion coefficient, e is the electron charge, n is the oxygen concentration in the oxide, and *k_B_* is the Boltzmann constant. Recalculation reveals that the oxygen-ion conductivity at 800 °C and 10 mbar is 0.0164 S/cm; and the total conductivity under identical conditions is 2.12 S/cm, according to E. Niwa et al. [44], i.e., two orders of magnitude higher.

The oxygen exchange coefficient differs significantly from that reported in the literature; a comprehensive plot is shown in Figure 6. The processes occurring at the surface can be highly dependent on the chemical and phase composition of the near-surface region of the material, which appears to be related to this discrepancy. According to [34], data obtained by the SIMS method exhibit substantial variation. For example, there is a deviation of 2.5 orders of magnitude for the surface exchange coefficient at 500 °C and an oxygen partial pressure of 20 mbar, as shown in Figure 6a (the marked area). In the opinion of the authors, the significant variation is due to the heterogeneity of the surface, specifically the presence of a nickel-enriched phase that has a negative effect on the kinetic characteristics. Since in the SIMS method, the depth of a small region determines the ^18^O concentration, the kinetic profile can be significantly distorted by heterogeneity. The EDX analysis of our case showed no changes in the phase composition of the surface, so it can be concluded that the chemical composition and phase are stable. Repeated kinetic experiments under identical conditions confirmed the results with good agreement. A strong correlation between the activation energies of the *k* process (0.48 ± 0.03 eV) and the SIMS data (0.57 ± 0.13 eV) is observed within the temperature range of 500–800 °C.

According to Budiman R. A. et al. [35], the oxygen exchange rate increases ten times when the oxygen pressure is raised from 10 to 100 mbar at 800 °C (Figure 6b). A similar trend in the increase in the surface exchange coefficient was observed in our experiment; namely, *k* changed from 6.58 × 10^−8^ cm/s to 1.07 × 10^−7^ cm/s at 800 °C with a pressure change from 10 to 20 mbar. Upon comparing data obtained from two different methods, IE-GPE and PIE, it can be seen that a pressure change from 20 to 200 mbar increases the surface exchange coefficient by one order of magnitude, i.e., to 1.63 × 10^−6^ cm/s when extrapolating the data to 800 °C. The calculation of the $\frac{logk}{logP_{O_{2}}}$ slope demonstrates a similarity of 1.01 ± 0.09 to [35]. Often, at high temperatures, the exchange occurs by the dissociative adsorption–desorption mechanism involving oxygen. This process includes a series of steps, and different values of $\frac{logk}{logP_{O_{2}}}$ slopes are observed, depending on the rate of the slowest of these steps.

A comprehensive analysis of the oxygen exchange mechanism can be found in the work of Fleig J. et al. [45], where there are several cases of oxygen adsorption particles and oxygen coverage of the oxide surface. In the case of oxygen reduction, the charge transfer process can be rapid in LNF oxide due to the high electronic conductivity, so oxygen exchange reactions involving $O_{ad}^{-}$, $O_{ad}^{2-}$, $O_{2,\;ad}^{-}$, or $O_{2,\;ad}^{2-}$ particles can be expected to take place. For these reactions, the slope of the carrier concentration versus oxygen pressure can be expressed as $\frac{logk}{logP_{O_{2}}}=1$. The transfer of oxygen ions may potentially be a limiting step in the oxygen reduction process due to the relatively low ionic conductivity. Surface processes involve multiple steps, and to identify the rate-determining steps, it is necessary to take into account the contributions of the three types of oxygen exchange, as well as the individual steps involved in dissociative adsorption and oxygen incorporation.

Figure 7a presents the isothermal dependence of the rates of the three types at different oxygen pressures. Measurements for ceramic materials are presented at oxygen pressures of 10 and 20 mbar. At these conditions, all three types of oxygen molecular interaction are present, with the *r*_0_ type being the dominant one. This exchange type is characterized by mechanisms that can occur via the adsorption–desorption process, when the rate of adsorption or desorption significantly surpasses the exchange rate of adsorbed atoms or ions with oxygen present in the outermost oxide layer and the oxide bulk. With an increase in oxygen pressure, there is an increase in exchange contributions that involve oxide lattice ions, specifically the rates of *r*_1_ and *r*_2_. Only the *r*_2_ type was found in the case of LNF powder at high oxygen pressures of 200 mbar. The increase in *r*_2_ type contribution with *P*_O_2__ and the decrease in other types of contribution can be attributed to two major factors: the change in the concentration and/or energy of surface adsorption centers, i.e., the surface homogeneity and the exchangeability of bulk oxygen governed by LNF morphology. The following section will discuss the contribution of these factors in more detail.

To assess the effect of surface homogeneity on surface oxygen exchange kinetics over dense ceramic and powder, it is necessary to calculate the surface inhomogeneity parameter, expressed as $P_{N}=\frac{2\sqrt{r_{0}r_{2}}}{r_{1}}$. As stated in [46], for single-phase oxide materials, *P_N_* can have values ≥ 1. These values correspond to the two types of oxygen surface exchange mechanisms: oxygen dissociative adsorption and incorporation. The two-step model proposed by M.W. den Otter et al. [47] and the statistical model proposed by Ananyev M.V. et al. [46] describe these mechanisms. For the two-step model, when *P_N_* = 1, all adsorption centers on the surface of a solid are considered to be equal and have the same energy, essentially indicating the homogeneity of the solid surface. For the statistical model *P_N_* >> 1 (or ∞ in the limit), elementary step rates related to oxygen adsorption and incorporation centers with different energy levels can be estimated (in the case of an inhomogeneous surface).

For the surface of LNF ceramics, the inhomogeneity parameter *P_N_* is equal to 1.47 ± 0.49 in the temperature range of 600–800 °C, and for LNF powder, *P_N_* is equal to 1, as its exchange kinetics is described with the pure *r*_2_ type. The *P_N_* values for both dense and powder LNF indicate that the mechanism of oxygen surface exchange for both is two-step and that the LNF surface is mostly homogeneous. Thus, the differences in the rates of the three types for dense and powder can be attributed to the morphological features of the investigated material. In order to explain the kinetic parameters and the influence of LNF morphology, the dissociative adsorption and incorporation rates can be calculated, and their values can be compared using the two-step model.

The elementary reactions for the oxygen dissociative adsorption and oxygen incorporation steps follow the two-step model. Rates *r_a_* and *r_i_* can be written as:(13)O16O18+O16a⇔raO162+O18a  or O182+O16a⇔raO16O18+O18a,
(14)O18a+O16S⇔riO18s+O16a,
where *O_s_* is the oxygen in the crystalline lattice of the oxide in the outermost layer and *O_a_* is the adsorbed oxygen in the adsorption center on the surface.

These rates for LNF material were determined for the first time, which allowed us to clearly identify the rate-determining step of the oxygen surface exchange process. The *r_a_* values for ceramic material and powder are similar, considering the higher *P*_O_2__ for the powder sample (Figure 7b). However, the apparent activation energies of dissociative adsorption differ, the most favorable one being 0.73 ± 0.05 eV on LNF powder, the energy barrier is higher on ceramic material (0.83 ± 0.06 eV and 0.98 ± 0.08 eV) and depends on the oxygen pressure (Table 2). The near-surface layers of the material may differ in the concentration of the elements and the ratio of the valence states of the transition metals. For example, Chen Zh. et al. [48] demonstrated through an XPS spectra analysis that the valence states of Fe^3+^/Fe^2+^ and Ni^3+^/Ni^2+^ ions, together with the correlation between *O_a_*/*O_s_* lattice oxygen and the adsorbed oxygen, differed in the near-surface layers, depending on the preparation method and morphology of LNF material. The total valence state of transition metals in the lowest oxidation state and the fraction of adsorbed oxygen are higher for the highly dispersed material compared to the dense material. This indicates that the former has greater oxygen adsorption capability. The difference in activation barriers for dissociative adsorption is likely due to variations in the valence states of iron and nickel ions, the degrees of surface coverage by adsorbed oxygen at different temperatures, and oxygen pressures for the ceramic and LNF powders.

There are differences in the oxygen incorporation rates measured by the PIE and IE-GPE methods. Figure 7b shows the *r_i_* values for the dense LNF plates obtained by the IE-GPE method. The *r_i_* values are lower than *r_a_* within the temperature range of 600–800 °C, and the activation energy is 0.42 ± 0.04 eV. The *r_i_* values measured for the powder LNF by the PIE method are excessively high, which prevents an accurate calculation of *r_i_* within the temperature range of 350–550 °C. It can only be said that according to PIE accuracy restrictions, the values of *r_i_* for LNF powder are at least two orders of magnitude higher than the *r_a_* values. The significantly different *r_i_* values, with relatively close *r_a_* values, support our previous conclusion that the main factor governing the oxygen surface exchange mechanism for LNF is the oxide morphology. The effect of sample morphology on the oxygen incorporation step is so pronounced that the rate-determining step of the oxygen surface exchange in the case of the dense LNF is oxygen incorporation into the oxide structure, while in the case of LNF powder, the rate-determining step of the oxygen surface exchange is the oxygen dissociative adsorption.

The influence of LNF morphology on the rate of oxygen incorporation can be caused by diffusion processes occurring in the dense material. According to Table 2, the apparent activation energy for the oxygen diffusion is higher than the oxygen incorporation by approximately 0.2 eV. This implies that ^18^O leaping between vacancies in the dense LNF is a less energetically favorable process than the oxygen incorporation from the surface oxide to the near-surface layer. Moreover, the size of powder particles in the powder LNF is much smaller than the size of the pellet in the case of the dense sample, which makes it easier for ^18^O to uniformly distribute in the sample bulk. Thus, the drastic difference in the *r_i_* rates measured for the dense and powder LNF is likely to be caused by diffusion limitation of the bulk ^18^O distribution in the dense LNF sample.

## 3. Materials and Methods

LaNi_0.6_Fe_0.4_O_3−*δ*_ powder was synthesized using the Pechini technique [27]. The nitrate salts La(NO_3_)_3_ × 6H_2_O, Ni(NO_3_)_2_ × 6H_2_O, and Fe(NO_3_)_3_ × 9H_2_O were used as initial components. The purity of the salts was at least 99%. Citric acid (99.8%) and ethylene glycol (99.6%) were used as organic fuel for complexation and combustion, respectively. The amount of starting salts per gram of final product is 4.09 mmol of La(NO_3_)_3_ × 6H_2_O, 2.45 mmol of Ni(NO_3_)_2_ × 6H_2_O, and 1.63 mmol of Fe(NO_3_)_3_ × 9H_2_O, respectively. Mixed components were continuously stirred on a heating plate (ULAB Scientific Instruments Co., Nanjing, China) in a glass vessel. Citrate complexes were obtained from starting component salts in excess of citric acid (1:1.5) by heating the mixture to 100 °C and through a reaction (15). Once the metal citrate complexes were formed, ethylene glycol (1:1) was added to the reaction mixture to form ester metal complexes (reaction 16), where Me denotes lanthanum, nickel, or iron. The reaction mixture was heated to 450 °C at a rate of 200 °C per hour to evaporate the solution without boiling it. The mixture took approximately 5–6 h to evaporate, after which it self-ignited. Upon ignition of the mixture, a fine powder was generated, which underwent rapid cooling and grinding. The precursor powder was heated to 900 °C and annealed for 6 h using a heating and cooling rate of 100 °C per hour to produce a single-phase material.
(15)$${\mathrm{Me}}^{n+}+C_{6}H_{8}O_{7}\to {\left[\mathrm{Me}\left(C_{6}H_{8}O_{7}\right)\right]}^{n+},$$
(16)$${\left[\mathrm{Me}\left(C_{6}H_{8}O_{7}\right)\right]}^{n+}+C_{2}H_{4}{\left(\mathrm{OH}\right)}_{2}\to {\left[\mathrm{Me}\left(C_{6}H_{7}O_{7}\right)-C_{2}H_{4}\left(\mathrm{OH}\right)\right]}^{n+}+H_{2}O.$$

X-ray powder diffraction (XRD) was performed using a Rigaku MiniFlex 600 diffractometer with a D/teX Ultra2 detector (Rigaku Co. Ltd., Takatsuki, Japan) in CuK*α* radiation at *λ* = 1.5405 Å; the measurements were performed at room temperature in air. The diffraction patterns were recorded with a scan step of 0.01 and a scan rate of 0.3/min at 2*Θ* 20–90°.

The specific surface area was determined by the BET method using Sorbi N.4.1 software (Meta, Novosibirsk, Russia). Prior to measurements, the samples were degassed for 1 h in a stream of 99.995% pure helium at 200 °C. Powder particle size analysis was performed using a laser analyzer, specifically the Malvern Mastersizer 2000 (Malvern, Worcestershire, UK). Distilled water-based powder suspensions were measured using optical radiation recorded with a He-Ne laser. The suspension was stirred with a 2600 rpm stirrer to break up agglomerated particles and was also treated with ultrasound.

LNF powder was ground and isostatically pressed at a pressure of 2 ton/cm^2^ to form a circular shape with a diameter of about 12 mm and a thickness of about 2 mm. The samples were heated at a temperature of 1350 °C in an air atmosphere, with a heating and cooling rate of 100 °C per hour, and held for 4 h. The ceramic material demonstrates a density that is 97% of the theoretical density derived from the crystallographic parameters. The samples were thinned to a plate form with a thickness of 1 mm. The plates were then polished using diamond paste with gradually decreasing grain size, starting from ASM 7/5 NVM (grain size 5–7 µm) and then ASM1/0 NOM (grain size 1 µm), and were subsequently cleaned by alternating ultrasonic baths of acetone, isopropyl alcohol, and deionized water.

The surface morphology and elemental mapping of the polished surface were examined by means of scanning electron microscopy using Mira 3 LMU (TESCAN, Brno, Czech Republic), which was equipped with an energy-dispersive X-act spectrometer (Oxford Instruments, UK).

Two independent oxygen isotope exchange methods were used to investigate in situ the interaction kinetics of oxygen from the gas phase with the samples. The powder material was investigated by pulsed isotope exchange (PIE) using the setup described in [49]. PIE experiments were performed in the temperature range of 200–900 °C under a continuous flow of a He + ^16^O_2_ gas carrier mixture with 21 vol.% of O_2_ (*P*_O_2__ = 200 mbar). The purities of He and O_2_ were 99.9999% and 99.999% (natural isotope composition), respectively. The temperature range was chosen experimentally so that at the lowest temperature the sample was inactive for oxygen isotope exchange. In order to obtain data at different times of pulse interaction with LNF, the measurements were performed at two different flow rates of the gas carrier mixture, 3.6 and 4 L/h. The 1 and 0.5 mL pulses of the N_2_ + ^18^O_2_ mixture with the same volume % O_2_ as in the gas carrier were injected during the PIE experiments at each gas carrier flow. The N_2_ gas in the pulse mixture was chosen as an internal standard due to its inactivity. The purities of N_2_ and ^18^O_2_ were 99.999 vol.% and 99.999 vol.% (96.4 at.% ^18^O enrichment), respectively. A Microvision 2 Vision 2000P residual gas analyzer (MKS Instruments, Andover, Massachusetts, USA) equipped with a multiplier detector was used for continuous monitoring of the gas phase composition during the PIE experiments. The calculation of the ^16^O_2_, ^16^O^18^O, and ^18^O_2_ mole fractions from the PIE results was carried out from the integral areas under the pulse peaks related to the 28, 34, and 36 masses (^14^N_2_, ^16^O^18^O, ^18^O_2_) according to the method presented in [49,50].

Dense ceramic materials have been investigated using a static circulation rig in which an oxygen isotope exchange with a gas phase equilibration (IE-GPE) technique has been implemented. Isotope experiments were performed on LNF plates in the temperature range of 600–800 °C and at oxygen pressures of 10–20 mbar. The purity of the oxygen used for the experiments was 99.999 vol% and the enrichment of ^18^O_2_ was 83.6 at.%. During the experiment, the time course of the ionic current corresponding to the masses 32, 34, and 36 (^16^O_2_, ^16^O^18^O, ^18^O_2_) was recorded using an Agilent 5973N quadrupole mass spectrometer (Agilent Technologies Inc., Santa Clara, CA, USA).

## 4. Conclusions

The measurements for the oxygen diffusion coefficient, the surface exchange coefficient, the dissociative adsorption, and the incorporation rates of LaNi_0.6_Fe_0.4_O_3−*δ*_ were obtained using the pulse oxygen isotope exchange and the oxygen isotope exchange with gas phase analysis in a temperature range of 350–800 °C and oxygen pressure of 10, 20, and 200 mbar. Employing two independent methods significantly increased the range of conditions, encompassing various temperatures and oxygen pressures. This allowed us to clearly demonstrate the influence of the external parameters and morphological features on kinetic parameters. Likely, the differences observed in the mechanism of oxygen surface exchange and diffusion on dense and powder samples are connected with changes in the microstructure and surface morphology of the samples.

This study demonstrates the effect of oxygen pressure on the ratio of the contributions of the three exchange types to the total oxygen exchange rate in LNF. At oxygen pressures of 10 and 20 mbar, the ceramic material exhibits all three types of oxygen molecule interactions, with the *r*_0_ type being the most prevalent. The contributions from the oxide lattice ions, i.e., the *r*_1_ and *r*_2_ rates, increase with an increase in oxygen pressure. Only the *r*_2_ type was detected in LNF powder at high oxygen pressures of 200 mbar. It is possible that the increase in the contribution of the *r*_2_ type with *P*_O_2__ and the decrease in the contribution of the other types are associated with changes in the concentration and energy of the surface adsorption sites, as well as the exchange capacity of the lattice oxygen.

The rates of dissociative adsorption and incorporation were determined for the first time. The incorporation rate is about one order of magnitude lower than the adsorption rate on a ceramic sample. Thus, at 800 °C and a pressure of 10 mbar, the oxygen adsorption rate is 6.68 × 10^16^ atom × cm^−2^ × s^−1^, and the oxygen incorporation rate is 6.59 × 10^15^ atom × cm^−2^ × s^−1^. For powder materials, the rate determinant is the dissociative adsorption rate. The influence of morphology on the rate ratio of the individual steps can be due to the diffusion processes that take place in the dense material.

The oxygen diffusion coefficient increases with rising temperatures and decreases with an increase in oxygen pressure. At 10 mbar and 800 °C, the oxygen-ionic conductivity calculated from the diffusion coefficient is 0.0164 S/cm, which is two orders of magnitude lower than the total conductivity of this material, 2.12 S/cm.

## Figures and Tables

**Figure 1 ijms-24-13013-f001:**
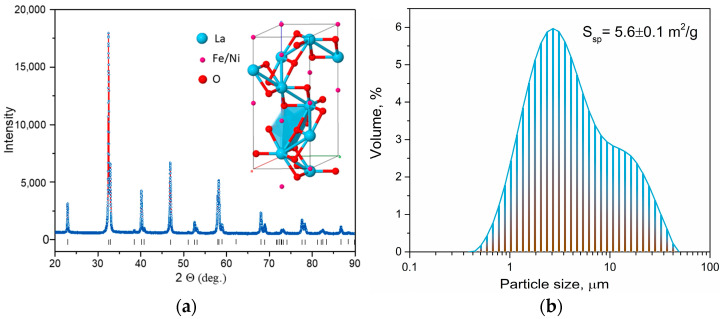
X-ray diffraction pattern (dots and line), where the vertical bars are the Bragg positions for the LaNi_0.6_Fe_0.4_O_3−*δ*_ crystal structure. The insertion corresponds to the structure LaNi_0.6_Fe_0.4_O_3−*δ*_: https://materials.springer.com/isp/crystallographic/docs/sd_1044291, 17 July 2023. (**a**) and volume particle size distribution (**b**) of LNF powder.

**Figure 2 ijms-24-13013-f002:**
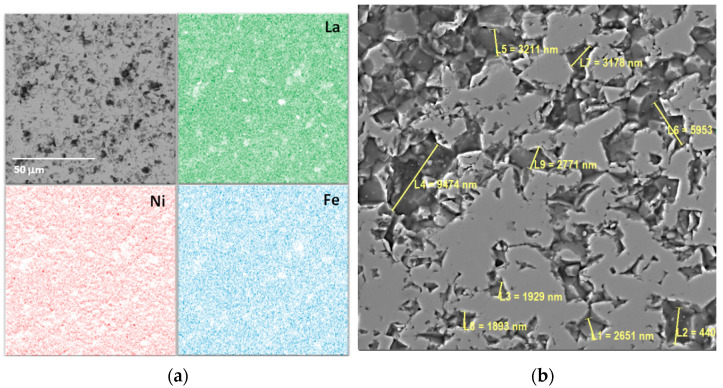
SEM image with EDX mapping analysis (**a**) and BSE image (**b**) of LNF plate.

**Figure 3 ijms-24-13013-f003:**
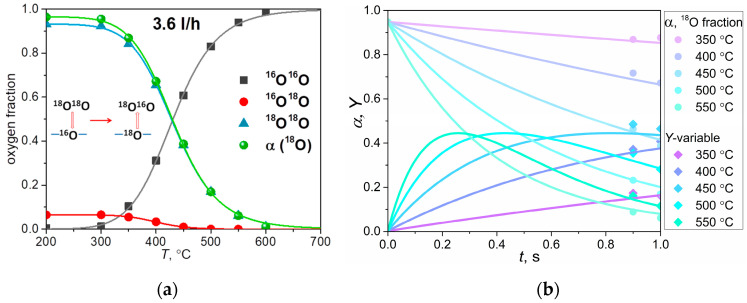
Temperature dependence of the oxygen isotopologue fractions (*x*_32_, *x*_34_, *x*_36_) and the ^18^O fraction (*α*) in the gas phase at 3.6 L/h gas carrier flow (**a**), in the insert is a schematic mechanism of oxygen exchange between O_2_ and surface; interaction time dependencies of the ^18^O fraction and the *Y* variable at 350–550 °C (**b**) obtained in the PIE experiments.

**Figure 4 ijms-24-13013-f004:**
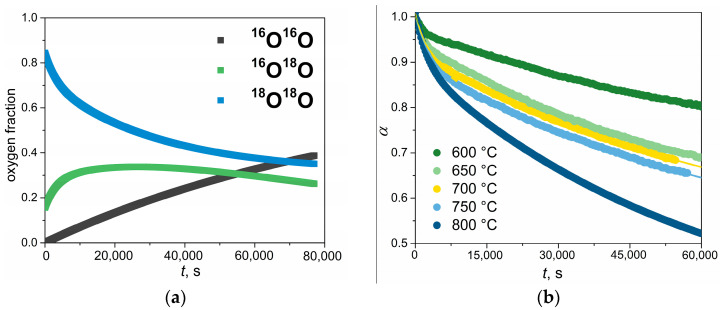
The time dependencies of the oxygen isotopologues fraction at 800 °C (**a**) and the ^18^O fraction (*α*) at 600–800 °C (**b**) obtained in the IE-GPE experiments at 10 mbar.

**Figure 5 ijms-24-13013-f005:**
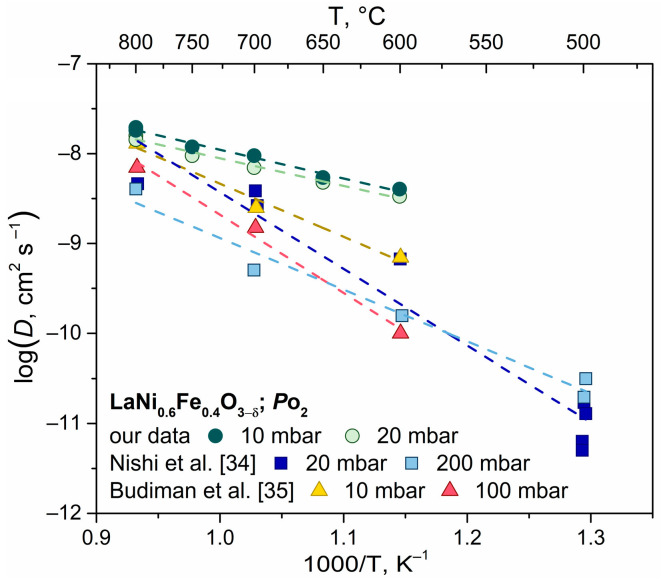
Arrhenius-type plot of the oxygen diffusion coefficient of LNF oxide measuring by different isotopic methods [34,35].

**Figure 6 ijms-24-13013-f006:**
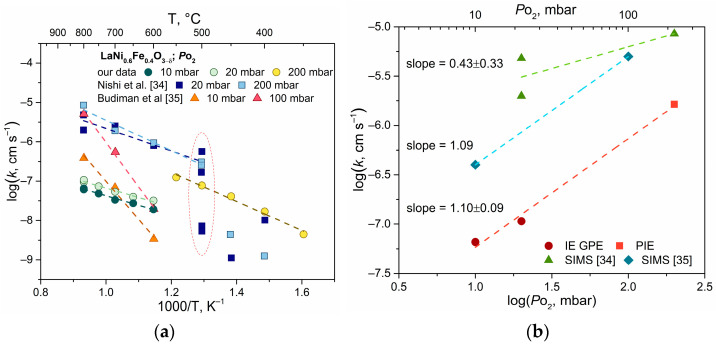
Arrhenius-type plot (**a**) and *P*_O_2__-type plot at 800 °C (**b**) of the surface exchange coefficient of LNF oxide measured by different isotopic methods [34,35].

**Figure 7 ijms-24-13013-f007:**
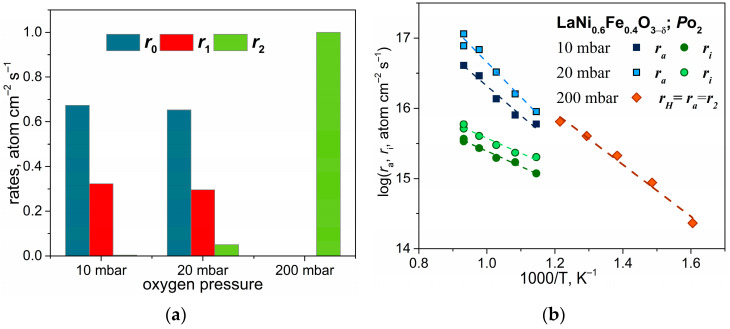
Contributions of the three types of oxygen exchange from pressure at 600 °C (**a**); temperature dependencies of dissociative adsorption and incorporation rates at different oxygen pressures (**b**).

**Table 1 ijms-24-13013-t001:** Element contents (atom%) and stoichiometric ratios on LNF surface.

Position	La	Fe	Ni	O
Segment I	21.51	8.21	12.95	59.01
Segment II	21.80	8.19	12.78	58.96
Segment III	20.91	7.80	13.21	58.58
Standard stoichiometric ratio	1.00	0.40	0.60	3.00
Stoichiometric ratio calculated as La equal to 1	1.00	0.38	0.62	2.75

**Table 2 ijms-24-13013-t002:** Apparent activation energies of the surface exchange, the oxygen diffusion, dissociative adsorption, and incorporation rates with respective experimental conditions for LaNi_0.6_Fe_0.4_O_3−*δ*_.

Method	*P*_O_2__, mbar	*T*, °C	Energy Activation, eV				Reference
			k	D	r_a_	r_i_	
PIE *	200	350–550	0.73 ± 0.05	–	0.73 ± 0.05	–	our data
IE-GPE *	10	600–800	0.48 ± 0.02	0.63 ± 0.04	0.83 ± 0.06	0.43 ± 0.03	our data
IE-GPE	20	600–800	0.48 ± 0.03	0.61 ± 0.04	0.98 ± 0.08	0.42 ± 0.04	our data
SIMS *	20	500–800	0.57 ± 0.13	1.69 ± 0.18	–	–	[34]
SIMS	200	500–800	0.76 ± 0.07	1.14 ± 0.11	–	–	[34]
SIMS	10	600–800	1.92 ± 0.17	1.17 ± 0.15	–	–	[35]
SIMS	100	600–800	2.24 ± 0.12	1.73 ± 0.17	–	–	[35]

* PIE—the pulse isotopic exchange technique, IE-GPE—the isotope exchange with the gas phase equilibration, SIMS—the secondary ion mass spectrometry.

## Data Availability

The data presented in this study are available upon request from the corresponding author.

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
