# Peer review of "Determination of Kinetic Parameters and Identification of the Rate-Determining Steps in the Oxygen Exchange Process for LaNi0.6Fe0.4O3−δ"

_ijms, 2023, doi:10.3390/ijms241613013_

Round 1

Reviewer 1 Report

First of all, I would like to thank you very much for choosing our journal for your article. It is a very meticulous and successfully prepared article. I would like to review the article again after making the corrections I mentioned.

While the overall Pechini process is explained, could you provide more explicit details about the reaction conditions such as the duration of heating, the temperature during the synthesis, and the molar ratio of the initial components?

Could you provide more information on the other characterization techniques used to evaluate the synthesized LaNi0.6Fe0.4O3–δ (LNF) powder, aside from XRD and BET? For instance, were techniques such as SEM or TEM used to study the morphology and size of the particles?

What are the potential applications for the LaNi0.6Fe0.4O3–δ (LNF) powder synthesized through this process? Are there any implications for its use in particular fields or industries?

Could you elaborate more on how the values for α and Y are derived and how they contribute to the understanding of oxygen isotope exchange in this context?

You compare two methods, PIE and IE-GPE. Could you elaborate on the advantages and limitations of each method, and why they are used in conjunction?

Can you explain the significance of the isotope exchange rates and oxygen diffusion coefficient values you obtained? How do they contribute to our understanding of the processes you are investigating?

You've mentioned that the results from this study differ from literature data, especially at lower temperatures. Can you explain why you believe this discrepancy exists? Is there anything about your experimental design or materials that could account for these differences? 

Can you elaborate on the slowing of diffusion transport observed with a pressure change from 10 to 20 mbar? Do you have a theoretical explanation or a model that explains this behavior?

How might these findings regarding oxygen diffusion and ionic conductivity in LNF oxide impact its potential applications? What might be the next steps in your research?

Your findings suggest that discrepancies in oxygen diffusion coefficient values might be attributed to grain boundary diffusion effects. Can you provide more explanation on how you ascertained this? Do you have any direct evidence or did you make any specific tests to prove this hypothesis?

You found that the oxygen-ion conductivity and the total conductivity at 800 °C and 10 mbar differed by two orders of magnitude. Can you explain why there's such a significant discrepancy?

You have mentioned a considerable variation in the data obtained by SIMS. What are the possible reasons for such a wide deviation? How does this variation influence your overall interpretation of your findings?

Based on this passage, you seem to be discussing the influence of surface homogeneity on the kinetics of the oxygen surface exchange over dense and powder LNF materials. You calculated a surface inhomogeneity parameter (?) which gives insight into the mechanisms of oxygen dissociative adsorption and incorporation on these surfaces.

You mentioned two types of mechanisms for oxygen surface exchange, each corresponding to certain values of ?: the two-step model and the statistical model. If I understood correctly, the two-step model (? = 1) indicates that all adsorption centers on the solid surface are equal and have the same energy, signifying surface homogeneity. In the statistical model, a high ? value (? >> 1) signifies an inhomogeneous surface with oxygen adsorption and incorporation centers of differing energies.

For your LNF materials, you found ? = 1.47 ± 0.49 for the ceramic surface and ? = 1 for the powder, both of which point towards a two-step oxygen exchange mechanism. This suggests that the surfaces of both the dense and powder LNF are mostly homogeneous.

In light of these results, you conclude that differences in the rates of three types for dense and powder LNF can be attributed to morphological features of the materials. Could you please elaborate on what these morphological features are, and how they might impact the oxygen surface exchange rates?

Reviewer 2 Report

The manuscript describes the LaNi0.6Fe0.4O3–δ (LNF) for oxygen exchange mechanism between O2 and LNF oxide which is a good topic and falls in the topic of the journal, however, there are some issues to be addressed. The comments are listed below:

1. The English of the text should be checked

2. The authors must be included new, relevant, and more information about other materials. Diverse studies are growing attention for diverse uses as reported by the Awual group according to ScienceDirect. The authors need to indicate such points for a broad range of readers. Moreover, the authors need to cite high-impact articles to make the manuscript high-level. The following specific articles may take be noted in the revision stage of https://doi.org/10.1016/j.colsurfa.2023.131859; https://doi.org/10.1016/j.seppur.2023.124088; https://doi.org/10.1016/j.colsurfa.2023.131794

3. Comparison between the obtained results and measured in this study with other reported studies should be done and included for more clarity (indicate values not just the number of references).

4. A schematic mechanism describing the oxygen exchange mechanism between O2 and LNF oxide must be indicated and included (reactions, interactions, etc.)

5. Correct the References using the guide of the Journal. More Conclusions must be included with the best results, and values obtained.

The English language needs to check carefully in the revision stage because of many careless mistakes in many positions.

Reviewer 3 Report

The authors discussed the differences in the mechanism of oxygen exchange and diffusion process in the dense ceramic and powder sample of LaNi0.6Fe0.4O3–δ (LNF). They intensively investigated the kinetic parameters, such as diffusion coefficients and activation energy, and also provided a comprehensive comparison of their results with the reported values. The detailed investigation and analysis they provided would be a ground reference for the readers of IJMC. However, the manuscript should be more organized, and further information can be added. Several comments they may need to consider are given below. 

1. As mentioned in the general comment above, I suggest that the authors reexamine their manuscript. For example, in line 67, “LNF” is used without definition – it is defined in the method section (line 74) for the first time. And also, in line 183, Oa is not shown in the above equation. Similarly, in equations (6)–(8), the meaning of r0, r1, and r2 is not addressed, and ra and ri are not even mentioned. These become clear in the equations (12) and (13). Such organization makes the manuscript hard to follow. 

2. The analysis is based on the kinetic parameters whose determination relies on the equations they presented. The validity of the equations seems important to suggest the significance and reliability of their result. Some details of the equations can be added to provide intuition. 

3. (Technical question) In the PIE measurements, For 16O2, the He+O2 mixture was used, but the N2+O2 mixture was used for 18O2. Any specific reason for that (He vs. N2)? 

4. (Technical question) How did the authors arrive at PN=1 for powder? I understand the equation in line 330 was not applicable to the powder measurement. 

5. In line 377, the authors state, “One can only be said that according to PIE accuracy restrictions the values of ri for the powder LNF in ≥ 2 order(s) of magnitude higher than ra values”. How could such a conclusion be derived? It seems a very important argument in this study because they mention that oxygen dissociative adsorption is the rate-determining step for the powder. 

6. For the caption of Figure 1, the authors need to address what the red line, blue circles, and vertical bars stand for. Even for other figures, readers would appreciate more detailed captions.

Commas and articles are missing in many places. 

Somewhat inappropriate expressions such as “let’s” need to be revised. 

(Line 141) microanalysis, Figure 2 → microanalysis (Figure 2). 

(Line 242) the data at 800 °C seem missing (1000/T = 1.25, according to the x-axis of Figure 5)

(Line 392) lower → smaller

Round 2

Reviewer 2 Report

The authors revised the manuscript based on the remarks to improve the manuscript for the reader’s understanding.

Accept.

Reviewer 3 Report

The authors have well-addressed the questions and the suggested revision, thus enhancing the quality and clarity of their manuscript. Based on the through review and the comprehensive revisions undertaken, I recommend the acceptance of their manuscript for publication.